# Neural network kinetics for exploring diffusion multiplicity and chemical ordering in compositionally complex materials

Bin Xing[1,2], Timothy J. Rupert[1,2], Xiaoqing Pan [1,2] & Penghui Cao [1,2,3] ✉

Diffusion involving atom transport from one location to another governs many important processes and behaviors such as precipitation and phase nucleation. The inherent chemical complexity in compositionally complex materials poses challenges for modeling atomic diffusion and the resulting formation of chemically ordered structures. Here, we introduce a neural network kinetics (NNK) scheme that predicts and simulates diffusion-induced chemical and structural evolution in complex concentrated chemical environments. The framework is grounded on efficient on-lattice structure and chemistry representation combined with artificial neural networks, enabling precise prediction of all path-dependent migration barriers and individual atom jumps. To demonstrate the method, we study the temperature-dependent local chemical ordering in a refractory NbMoTa alloy and reveal a critical temperature at which the B2 order reaches a maximum. The atomic jump randomness map exhibits the highest diffusion heterogeneity (multiplicity) in the vicinity of this characteristic temperature, which is closely related to chemical ordering and B2 structure formation. The scalable NNK framework provides a promising new avenue to exploring diffusion-related properties in the vast compositional space within which extraordinary properties are hidden.

Diffusion in materials dictates the kinetics of precipitation[1], new phase formation[2] and microstructure evolution[3], and strongly influences mechanical and physical properties[4]. For example, altering nanoprecipitate size and dispersion by thermal processing enables substantial increases in strength and good ductility in multicomponent alloys[5,6]. Essentially rooted in diffusion kinetics, predicting how fast local composition and microstructure evolve is a fundamental goal of material science. In metals and alloys, diffusion processes are connected with vacancies, point defects that mediate atom jumps in the crystal lattice. Molecular dynamics (MD)[7] modeling based on force fields or density functional theory, which probe the atomic mechanisms of diffusion at a nanosecond timescale, are often not able to access slow diffusion kinetics-induced microstructure change. To circumvent this time limitation inherent in MD, the kinetic Monte Carlo method (kMC) is primarily adopted to model diffusion-mediated structure evolution, for instance, the early stage of precipitation in dilute alloys[8,9]. In the kMC simulations, the crucial parameter (vacancy migration energy) is generally parameterized from continuum models such as cluster expansion[10] or Ising model[11], owing to the high computational cost in transition state search. The rise of compositionally complex alloys (CCAs), commonly known as high-entropy alloys, brings many intriguing kinetics behaviors, ranging from chemical short-range ordering[12], precipitation[6], segregation[13], and radiation defect annihilation[14], which have yet to be fundamentally understood and ultimately predicted. The chemical complexity in CCAs, however, poses a new challenge for modeling diffusion-mediated processes due to local chemical fluctuations leading to diverse activation barriers (i.e., a wide spectrum)[15].

The emergence of machine learning methods has demonstrated the potential for addressing computationally complex problems in

[1]Center for Complex and Active Materials, University of California, Irvine, CA, USA. [2]Department of Material Science and Engineering, University of California, Irvine, CA, USA. [3]Department of Mechanical and Aerospace Engineering, University of California, Irvine, CA, USA. ✉e-mail: caoph@uci.edu

materials science that involve nonlinear interactions and massive combinatorial space[16]. One of the most promising examples is machine-learned interatomic potentials that map a three-dimensional (3D) atomic configuration to its conformational energy with a high accuracy at a substantially reduced computational cost[17]. The key step in machine learning in molecular science is converting atomistic structure into numerical values (descriptive parameters–descriptor[18]) to represent the individual local chemical and structural environments. Two successful atomic environment descriptors are atom-centered symmetry function[19] and smooth overlap of atomic position[20]. The dimension of these local structure descriptors (consideration of all neighboring atoms within a cutoff distance) increases quadratically with the number of constituent elements[21], which escalates the number of parameters and training time for the application of machine learning to chemically complex CCAs. To address this issue, active efforts have been taken to compress chemical information and reduce the size of representation of local atomic environment[22–24]. Using the structure descriptor, atomic site-related scalar values, such as segregation energy[25] and atomic propensity to rearrange[26], have been predicted through machine learning models. Concerning vacancy diffusion in compositionally complex alloys, a critical parameter of interest is diffusion energy barrier $\Delta E$, i.e., the energy difference between transition state and the initial energy minimum (Supplementary Fig. S1). Due to atomic-scale composition fluctuation and the existence of multiple diffusion directions in CCAs, it necessitates a machine learning model to precisely predict vectoral property, specifically, diffusion path-dependent barriers. Another complexity, needing to be addressed in modeling diffusion and new phase formation in CCAs, lies in the extensive compositional space and the development of local chemical order, both of which profoundly impact on diffusion barriers and kinetics.

In this study, we introduce a neural network kinetics (NNK) scheme for predicting atomic diffusion and its resulting microstructure evolution in compositionally complex materials. Grounded on an efficient on-lattice atomic representation that converts individual atoms to neurons while preserving the atomic structure, the NNK precisely describes atomic (interneuron) interactions through a neural network model and predicts neuron kinetics evolution, embodying physical atom diffusion and microstructure evolution. With only one-time conversion of atomic configuration to neuron map, vacancy diffusions and chemical evolution are simulated by swapping neurons, rending high efficiency and scalability. Using refractory NbMoTa as a model system, we explore chemical ordering and B2 phase formation mediated by diffusion kinetics and reveal the anomalous diffusion (diffusion multiplicity) that is inherent in CCAs.

## Results

### Neural network kinetics scheme

Figure 1a shows the on-lattice structure and chemistry representation, where the initial atomic configuration with a vacancy is encoded into a digital matrix, or neuron map. The digits (1, 2, and 3) represent the corresponding atom types, and 0 denotes the vacancy (refer to Supplementary Fig. S2 for conversion and visualization of 3D crystals). This digital matrix capturing structure and composition features offers several advantages important as a descriptor[18]. The map dimension $O(N)$ scales linearly with the number of atoms $N$ and is invariant to the number of constituent atom types, which has the lowest dimension possible as the descriptor. Unlike traditional descriptors, the neuron map not only reflects the local chemical environment of individual atoms but also, more significantly, captures the entire system. Importantly, the determination of the descriptive map is simple and involves no intensive calculation or painstaking parameter tuning. Essential for diffusion, the representation can be rotationally covariant and enables prediction of diffusion path-dependent activation barriers (vector quantities). These vectorized digits are then passed to the NNK model and serve as input neurons.

Figure 1b depicts the schematic of the NNK which consists of an artificial neural network and a neuron kinetics module. The introduced neural network (with more than two hidden layers) is designed to learn the nonlinear interactions between input neurons (i.e., atoms and vacancy), and to output the diffusion energy barriers. Notably, the network only uses the vacancy and its neighboring neurons as inputs, resulting in a low and constant computational cost (independent of system size) without sacrificing accuracy (see Supplementary Note 2 for details). With the available barriers associated with each individual diffusion path, the neuron kinetics module adopts the kinetic Monte Carlo method to carry out diffusion kinetics evolution (see "Methods"). There are two features rendering the NNK a high computational efficiency and scalability with system size. First, the descriptor map is calculated only once for the initial atomic configuration, because

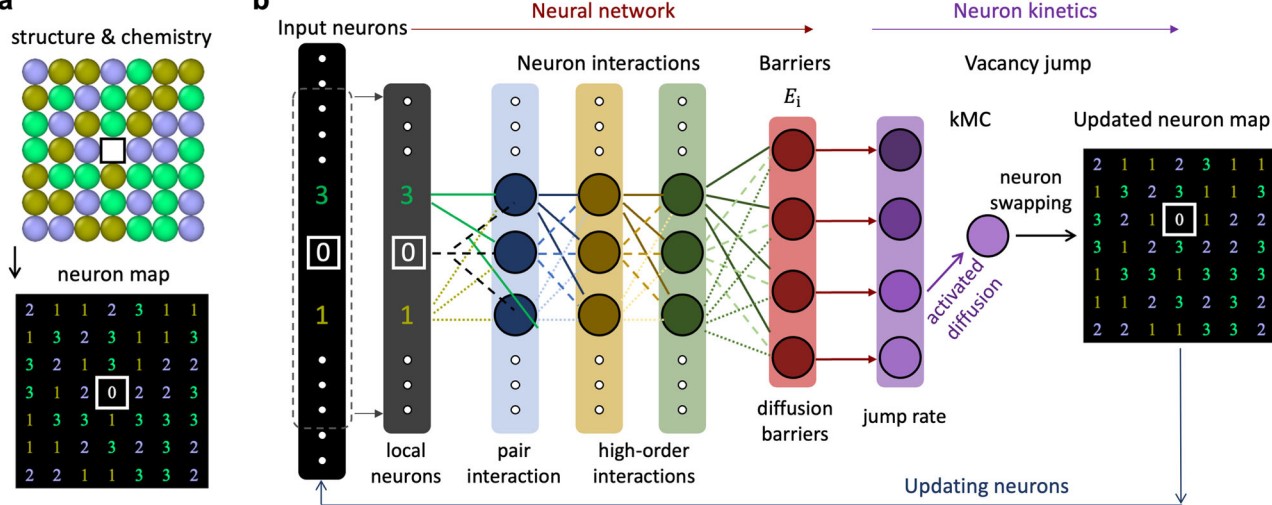

**Fig. 1 | Schematic illustration of neural network kinetics (NNK) framework.** **a** The on-lattice structure and chemistry representation of the entire system. A vacancy and its local atomic environment are encoded into a digital matrix (neuron map). **b** NNK framework consists of a neural network that outputs vacancy migration barriers, and a neuron kinetics module that implements neuron jump (diffusion jump) based on kinetic Monte Carlo (kMC). See "Methods" for details on neuron kinetics. Vacancy jumps and chemical evolution are efficiently modeled by swapping of neurons and neuron map evolution.

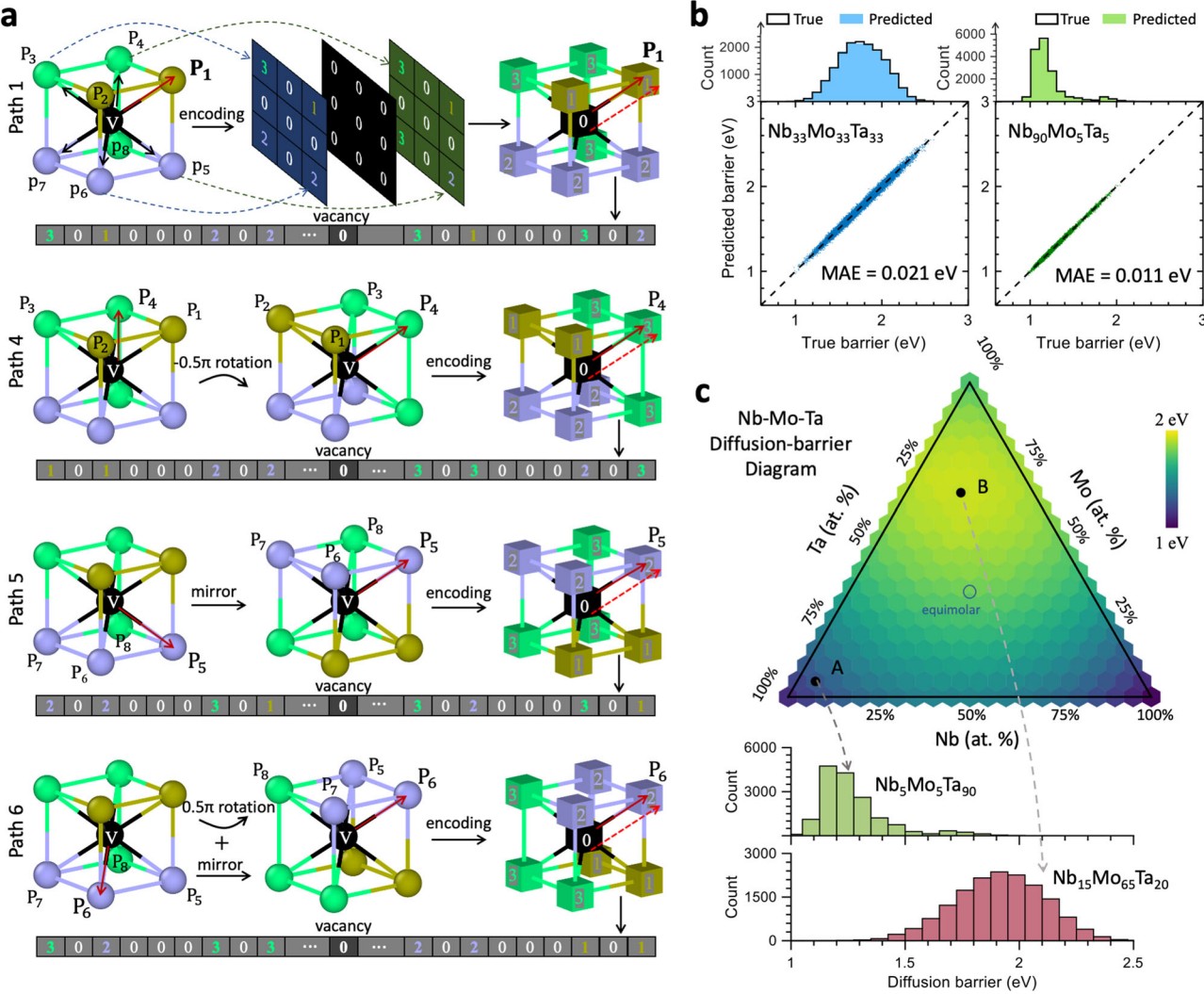

**Fig. 2 | Predicting diffusion barrier spectra in the entire composition space of Nb−Mo−Ta. a** Creation of unique neuron maps and feature vectors for each individual diffusion path P, which enables the prediction of eight path-dependent barriers from a vacancy. The symbol V represents the vacancy. **b** Performance of neural network in predicting diffusion barrier spectrum in concentrated, $Nb_{33}Mo_{33}Ta_{33}$, and dilute, $Nb_{90}Mo_5Ta_5$, solutions. **c** Diffusion barrier diagram generated by the neural network. The nonequimolar $Nb_{15}Mo_{65}Ta_{20}$ alloy exhibits the highest barrier in the Nb−Mo−Ta system.

atomic diffusion and local chemical evolution are operated on the representing neuron map. Second, since atomic diffusion depends solely on the local chemical environment, the NNK trained on small configurations can be directly applied to large systems for diffusion modeling. Therefore, with only a one-time conversion of atomic configuration to neuron map, vacancy jumps and chemical evolution can be simulated by swapping two digits of the neural map. In this way, millions of vacancy jumps can be modeled efficiently, with each jump iteration involving the action of just two neurons (Fig. 1b).

### Predicting a path-dependent diffusion barrier spectrum in multidimensional composition space

Diffusion in crystals occurs through elementary atomic jumps between a vacancy and its neighboring lattice sites (vacancy mechanism[4,27]). In body-centered cubic (bcc) CCAs, a vacancy is associated with eight different jump directions, and the variation in the jumping atoms and surrounding chemical environment can result in eight distinct migration barriers[15,28]. By utilizing the rotational covariance of lattice representation, it is possible to predict the jump path-dependent barriers (a vector quantity) from a single chemical configuration. Specifically, by aligning each diffusion path to a constant reference orientation through rotation and/or mirroring operations,

unique neuron map and digital vector, $D_i$, can be generated for each individual diffusion path $i$, without breaking the structural symmetry, as demonstrated in Fig. 2a. The Supplementary Table S1 and Supplementary Fig. S3 summarizes the operations aligning the diffusion direction of interest with this reference, preserving structural symmetry.

The neural network takes in $D_i$, which carries local atomic environment encompassing the vacancy, as input. The data (atomic digits) then flow through hidden layers to the output layer, which predicts the associated diffusion activation barrier, $E_i$. The first hidden layer in neural network characterizes the linear contribution of the input neurons (atoms and vacancy) to the migration barrier, while the following hidden layers capture the nonlinear and high-order interactions that impact vacancy jump. With just four hidden layers and 112 neighboring atoms (up to the 8th nearest neighbor shell) of the vacancy, the neural network achieves a high level of accuracy in predicting the path-dependent diffusion barrier (Supplementary Note 3 and Supplementary Figs. S12–14 for the testing of different neural network structures). Figure 2b presents the evaluation of machine learning model performance for two different concentrations (one concentrated and one dilute), where the predicted energy barrier value is compared with the ground truth (see "Methods"). The

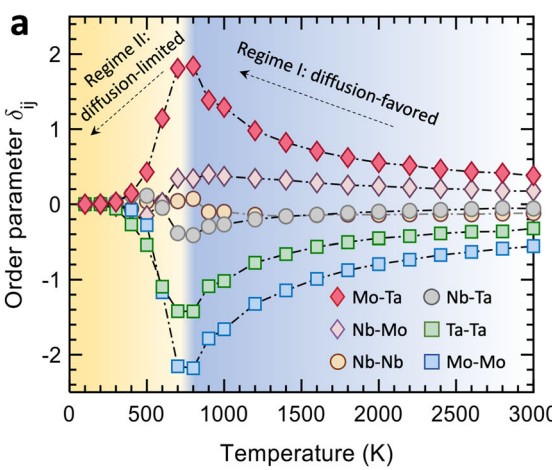
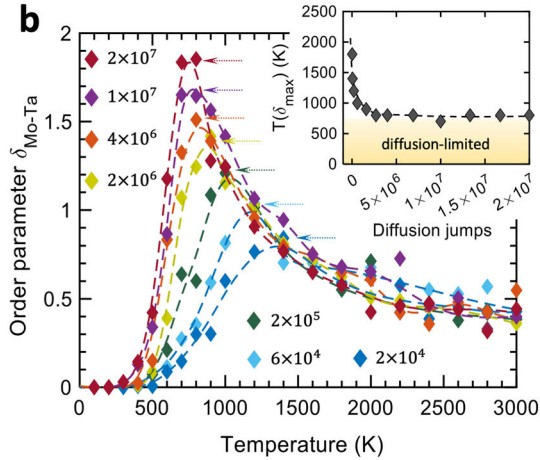

**Fig. 3 | Diffusion kinetics-mediated local chemical order in the equimolar NbMoTa alloy. a** Variation of chemical order $\delta_{ij}$ obtained at different annealing temperatures displays a critical temperature that divides the map into two characteristic regimes, denoted as diffusion-favored (I) and diffusion-limited (II).
**b** Development of Mo–Ta order, $\delta_{\text{Mo−Ta}}$, as a function of diffusion jumps from $2 \times 10^4$ to $2 \times 10^7$. The inset shows that the jump number dependence of peak temperature converges to the critical value ~800 K below which the chemical ordering is suppressed.

predicted and true values exhibit the same spectrum of barriers, and the mean absolute error (MAE) is less than 1.2% of the average true migration barrier for the two alloys, concentrated solution $Nb_{33}Mo_{33}Ta_{33}$ and dilute solution $Nb_{90}Mo_5Ta_5$ (see Supplementary Fig. S4 for new compositions and different system sizes).

After training on only tens of compositions (Supplementary Note 4 and Supplementary Figs. S15 and 16), the neural network remarkably harnesses the complete composition space of the ternary Nb–Mo–Ta system, building the relationship between composition and diffusion barrier spectrum. Figure 2c shows the diffusion barrier diagram generated by the neural network, from which the alloy ($Nb_{15}Mo_{65}Ta_{20}$) having the highest mean barrier is quickly identified. While research efforts have been primarily focused on equimolar or near-equimolar compositions, our results indicate an abnormal behavior can originate from nonequimolar concentrations hidden in the vast composition space. The neural network, which accurately predicts diffusion barriers for new and unseen compositions, implies that it fully deciphers the complex local chemistry variation and links it with diffusion property.

### Diffusion kinetics-induced local chemical order

Originating from attractive and repulsive interactions among the constituent elements of CCAs, atomic diffusion leads to the emergence of local chemical order on a short- to medium-range scale. To uncover diffusion-mediated chemical ordering and its dependence on annealing temperature, we employ the NNK model to performe aging simulations of the equimolar NbMoTa alloy at temperatures ranging from 100 to 3000 K. With the ability to resolve individual atomic jump and the low computational cost, 20 million diffusion jumps are carried out for each temperature.

Figure 3a shows the change of the local chemical order $\delta_{ij}$ as a function of temperature. Here the non-proportional order parameter, $\delta_{ij}$, quantifies the chemical order between a pair of atom types $i$ and $j$ in the first nearest neighbor shell (see "Methods"). A positive $\delta_{ij}$ indicates a higher number of pairs compared to a random solid solution, suggesting that element $i$ prefers to bond with element $j$ (favored pairing), while a negative value suggests an unfavored pairing. At a high temperature (3000 K), the system ultimately approaches the random solid solution, as reflected by the small value of $\delta_{ij}$. As the temperature decreases, the magnitudes of $\delta_{ij}$ for Mo–Ta, Ta–Ta, Mo–Mo pairs increase monotonically until they reach a turning point (around 800 K), beyond which the trend reverses. The chemical order falls rapidly as the temperature is lowered and, at 400 K, it nearly vanishes. It is noted that the system experienced an identical number of 20 million jumps at all temperatures.

These results suggest the existence of a critical temperature at which the diffusion-favored ordering reaches a maximum (Regime I in Fig. 3a). Below the critical temperature (Regime II), diffusion jumps barely develop and enhance chemical order.

To better understand this critical temperature and how the number of diffusion jumps affects it, we present the $\delta_{\text{Mo−Ta}}$ order parameter values obtained from a wide range of jumps, from $2 \times 10^4$ to $2 \times 10^7$, in Fig. 3b. As the number of diffusion jumps increases, the characteristic temperature $T(\delta_{\text{max}})$ corresponding to the maximum order gradually shifts to lower values and finally converges to 800 K. The inset of Fig. 3b illustrates the variation of $T(\delta_{\text{max}})$ with diffusion jumps, again unveiling this critical temperature below which diffusion-mediated ordering is substantially limited.

### Jump randomness and diffusion multiplicity in CCAs

In monoatomic crystals, the diffusion of vacancy can be described as purely random, with each possible jump path having an equal probability of occurrence. However, in CCAs, local variations in chemical composition give rise to distinct and path-dependent energy barriers, resulting in a multivariate distribution of jump probabilities. For example, in bcc CCAs, the jump probability for each of the eight possible paths associated with a vacancy site can be expressed as $p_i = \exp(-E_i/k_BT)/\sum_{j=1}^{8}\exp(E_j/k_BT)$, where $E_i$ is the energy barrier of path $i$, $k_B$ is Boltzmann constant, and $T$ is temperature. This can lead to various diffusion modes, as illustrated in Fig. 4a, where the two limiting jump cases are presented. One is pure random jump (where all jump paths have the same probability of occurrence), and the other is non-random, directional lattice jump (where one path predominates). To quantify the degree of lattice jump randomness, we define an order parameter $R = 1 - \sigma(\mathbf{p})/\max(\sigma)$, where $\sigma(\mathbf{p})$ is the standard deviation of jump probability, $\mathbf{p}$, and $\max(\sigma)$ is the maximum standard deviation occurring in directional or selective jump. Note the parameter, $R$, ranging from 0 to 1, quantifies the degree of jump randomness, with $R = 1$ and $R = 0$ representing the limiting cases of random diffusion and directional diffusion, respectively.

Figure 4b shows spatial and statistical distributions of lattice jump randomness $R$ at three representative temperatures. The spatial maps display color-coded lattices based on their respective $R$ values. At a high temperature of 3000 K, the thermal energy ($k_BT \gg E_i$) smears out the energy barrier difference between paths, leading to a peak $R$ value of 0.7, indicating highly random jumps. It is tempting to speculate that random atomic diffusion is insufficient to build and develop B2 ordered phase, which apparently corresponds to the low order

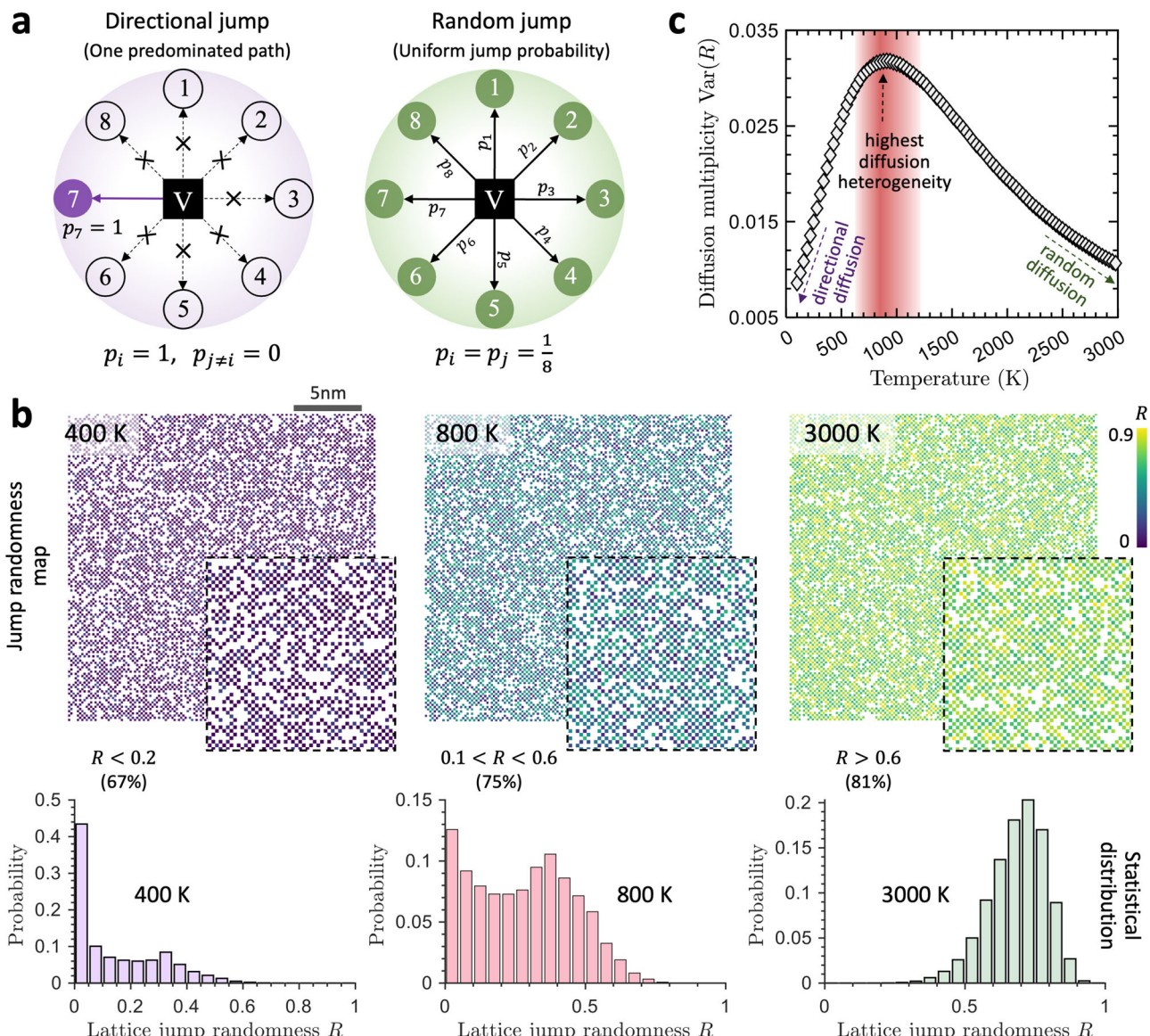

**Fig. 4 | Jump randomness and diffusion multiplicity of an equimolar NbMoTa alloy. a** Schematics of two limiting lattice jump modes. One of the eight paths is predominated in directional jump (jump randomness $R = 0$), while all eight paths have the same hopping probability $p$ in random jump ($R = 1$). **b** Spatial and statistical distributions of lattice jump randomness, $R$, at three representative temperatures. At 3000 K the distribution of $R$ ($R_{peak} = 0.7$) indicates highly random diffusion, while at 400 K the lattice jumps transform to directional (selective) diffusion mode ($R_{peak} = 0.0$). Lattice jumps at 800 K exhibit highly heterogeneous diffusion modes, shown by the broad distribution of $R$. **c** Diffusion multiplicity $Var(R)$ as a function of temperature reveals a critical temperature (~850 K) at which diffusion is more heterogeneous (widest distribution of $R$). Moving to the two ends, diffusion approaches simple random and directional modes at ultimate high- and low temperatures, respectively.

observed at high temperatures (Fig. 3a). At a low temperature of 400 K, the lattice jumps transform into directional diffusion, as demonstrated by the $R$ distribution having a peak value of 0. This implies that only one of the eight diffusion pathways is active at each lattice site. Presumably, this one-dimensional directional diffusion predominating at low temperatures (<400 K) limits and suppresses the nucleation and growth of three-dimensional B2 structure. Intriguingly, at an intermediate temperature (~800 K), the lattice jump randomness $R$ exhibits a broad distribution, spanning from 0.0 to 0.7, indicating highly heterogeneous diffusion modes.

To assess the system-level diffusion multiplicity (heterogeneity) and its temperature dependence, we calculate the variance of diffusion randomness $Var(R)$ across temperatures ranging from 100 to 3000 K, as illustrated in Fig. 4c. When close to the high or low-temperature ends, there is a rapid change in $Var(R)$, implying that diffusion approaches a random or directional mode. The temperature variation

of $Var(R)$ reveals a peak value of diffusion multiplicity at around 850 K. Random and directional-type lattice jumps are spatially interspersed throughout the entire system, as shown in the spatial map of Fig. 4b. The observation of the highest diffusion multiplicity (Fig. 4c) and maximum B2 order (Fig. 3a) occurring in the similar intermediate temperature range suggests a strong correlation between diffusion heterogeneity and the formation of B2 order.

## B2 structure nucleation and growth kinetics

Determining the formation kinetics of chemically ordered structure in a complex solid solution has been a challenge due to the local chemical fluctuations and huge amounts of diffusion barriers. The NNK framework efficiently and precisely predicting diffusion barrier at any chemical environment is intended to address this issue. To demonstrate the efficacy of the model, we perform aging simulations of NbMoTa consisting of 128,000 atoms. Figure 5a–c shows the spatial-temporal

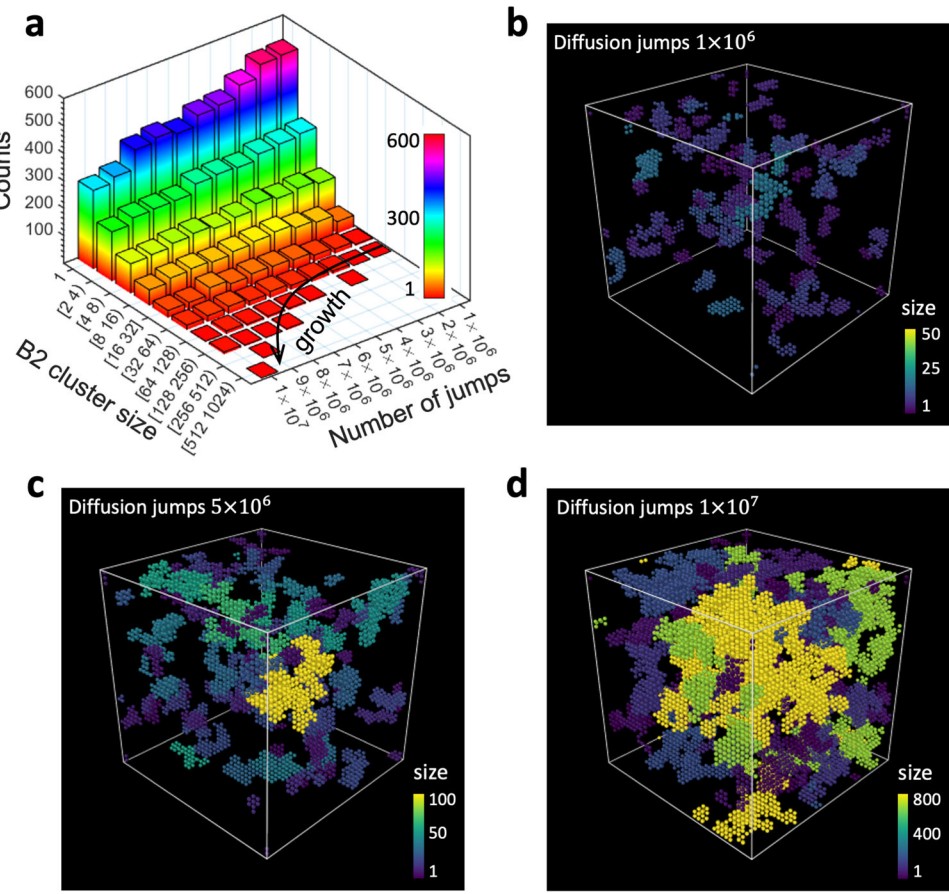

**Fig. 5 | B2 structure nucleation and growth kinetics during annealing in NbMoTa. a** B2 cluster size evolution with the number of diffusion jumps. **b–d** Spatial distributions of growing B2 cluster at $1\times10^6$, $5\times10^6$, and $1\times10^7$ diffusion jumps. Clusters are color-coded by their size.

nucleation and evolution of B2 structure induced by diffusion. With $1\times10^6$ diffusion jumps, a considerable amount of B2 clusters emerge in the system (Fig. 5b), most of which are small clusters (size <8 atoms). As the number of diffusion jumps further increases ($5\times10^6$), large clusters begin to appear and continue to grow, accompanied by annihilation and reduction of small ones (Fig. 5a). The decrease in spatially isolated small clusters are a result of their attachment or adsorption by nearby growing large ones. Apart from small clusters, another essential kinetic process underlying growth is large cluster interaction and coalescence. When two spreading clusters come near to each other, they merge into a large one mediated by diffusion (Supplementary Fig. S8). Figure 5d reveals the spatial distribution of formed B2 clusters colored by their size in the aged material. In contrast to the precipitation of ordered nanoparticles in dilute solutions, the more heterogenous growth of chemically ordered structure signifies the substantial role of diffusion multiplicity in governing the complex chemical ordering in concentrated solutions.

## Discussion

Diffusion kinetics in the emergent compositionally complex materials[29,30] (often called high-entropy alloys and high-entropy oxides) raise many intriguing rate-controlling phenomena and properties, such as chemical short-range order[12], chemically ordered nanoparticle formation[31], decomposition[32], superionic conductivity[33], extraordinary radiation tolerance[14,34], to new a few. These behaviors are controlled by the underlying atomic diffusion, which occurs in a chemical environment with a high degree of local composition fluctuations. Uncovering the kinetic processes and predicting structure evolution in these materials requires novel computational techniques that can disentangle their chemical complexity and connect it with individual atomic jumps. The

NNK scheme introduced here aims to tackle the kinetic behaviors arising from diffusion processes, with a particular focus on this novel class of materials. Underpinned by an interpretable chemistry and structure representation (neuron map), the neural network precisely predicts the diffusion path-dependent energy barriers governing individual atomic jumps. The atomic diffusion and structure variations are effectively modeled on the neuron map through neuron digit exchange (Fig. 1b). This framework possesses three key advantages that give both high computational efficiency and accuracy in modeling diffusion and new phase formation. First, the interpretable on-lattice representation, which converts chemistry and structure to physically equivalent neuron maps, yields an ultra-small feature size, critical for machine learning models. Second, the determination of neuron map (descriptor) is a one-time and simple process, as it can be updated to fully replicate atomic diffusion jumps and structure evolution. Importantly, the rotational covariance of the neuron map enables the prediction of vector values from a single-neuron map (vacancy configuration). Third, the NNK trained by small models can be applied directly to investigate the kinetic behavior of large systems without sacrificing accuracy. This size scalability is demonstrated, for instance, by accurate barrier predictions (see Supplementary Figs. S4 and S20) and ordered phase growth in large NbMoTa systems (Fig. 5).

Cluster expansion (CE)[10,35] method has long been used to study thermodynamic properties of multicomponent systems, such as vacancy formation energy[36]. For diffusion kinetics, the pivotal factor is determining the diffusion barriers, requiring calculation of transition states (saddle points). While the CE has been commonly employed to predict the energies of local minimum states[8,9], presenting the transition state using CE and predicting the associated energy barrier remains a challenging task[37] (Supplementary Note 5). Particularly, the

increase in chemical complexity makes the design of clusters for even local minimum configurations a time-consuming process. To tackle this challenge, an approach involving parametrizing the reaction coordinate and minimum energy path has been proposed[38], however, leading to a low prediction accuracy. Another machine learning model promising to atomistic modeling is graph neural network (GNN), which has shown great success in developing universal machine learning interatomic potentials[39,40]. Regarding vacancy diffusion in CCAs, GNN theoretically has the potential to predict vector properties using rotation-covariance features. However, modeling vacancy jump and chemical evolution using graph network entails node swapping and updating edge properties. Each node swap (representing vacancy jump) can potentially affect neighboring nodes and their connected edge features, necessitating their updates. This requires altering a significant portion of the network, encompassing the 8th nearest neighbors of vacancy node, after each vacancy jump. In contrast, the introduced NNK scheme, with neural map representation, simplifies the process by only requiring the update of two neurons for each diffusion jump. This simplicity allows the mirroring of vacancy jumps through the swapping of neurons (digits). With just one-time conversion of the atomic configuration to a neuron map, vacancy diffusion and chemical evolution are efficiently simulated by swapping digits (the vacancy neuron and one of its nearest-neighboring neurons) according to precise diffusion barriers and system temperature. In this way, tens of millions of vacancy jumps are modeled efficiently, with each jump iteration involving the action of just two neurons.

Stemming from attractive/repulsive interactions between solutes, atomic diffusion inevitably leads to nucleation of chemically ordered structure in CCAs during annealing. Using the NNK and bcc NbMoTa as model system, we uncover the existence of a critical temperature, at which the B2 order reaches its maximum value. This temperature dependence of chemical order is closely related to the underlying lattice jump randomness, as shown by the randomness maps (Fig. 4). At high temperatures close to the melting point, diffusion jumps ultimately approach a purely random process, corresponding to a low propensity for order formation. At low temperatures, lattice diffusion becomes dominated by the lowest barrier path, manifesting as directional jumping and restricting the nucleation of chemically ordered structure. At the critical temperature in the intermediate range, random-like and directional-type lattice jumps spread the entire system, exhibiting the highest diffusion heterogeneity (multiplicity, Fig. 4c). By tracking individual B2 clusters during annealing, it is found that their nucleation and growth are intermittent and non-uniform, accompanied by the reduction and annihilation of small clusters (Fig. 5 and Supplementary Video 1). This salient feature in the kinetics growth of B2 structure is not captured by fictitious thermodynamics-based modeling using random atom type swap (see "Methods" and Supplementary Fig. S9), which shows a more uniform growth (Supplementary Fig. S10). These results highlight the complex and multitudinous kinetic pathways in CCAs toward stable states, where many processes like ordered structure nucleation, annihilation, growth, and rearrangement are interplayed and coordinated.

The neural network trained on dozens of compositions demonstrates high performance for unseen compositions, unveiling the entire ternary space of Nb−Mo−Ta (Fig. 2c). With the design space for composition being practically limitless, the compositionally complex material formed by mixing multiple elements opens a new frontier waiting to be explored. Traditional structure-property calculations relying on density functional theory and molecular dynamics work well for small datasets but fall short in harnessing the vast composition space. Recent advances in the rapidly growing field of machine learning creates a fertile ground for computational material science[41,42], having led to the discovery of alloys with optimal properties[43]. By directly connecting the multidimensional composition with diffusion barrier spectra, the NNK illuminates a bright path to explore the vast compositional space of CCAs, where hidden extraordinary kinetic properties lie.

## Methods

### Material system and diffusion barrier calculation

We focus on the emergent refractory CCA, Nb−Mo−Ta, as the study system to demonstrate the neural network kinetics (NNK) scheme. When generating diffusion datasets for training the neural networks, we use atomic models consisting of 2000 atoms. To compute the vacancy diffusion energy barriers for the Nb−Mo−Ta system, we utilize the climbing image nudged elastic band (CI-NEB)[44] method and a machine learning potential[45]. In a bcc structure, vacancy jump has eight pathways and final configurations, which can be created by exchanging the vacancy with its nearest neighbor atoms. By labeling each jump path, the path-dependent energy barriers are calculated and stored for machine learning model training and validation. In the CI-NEB calculations, we set the inter-replica spring constant to $5.0\ eV/\text{Å}^2$. The energy and force tolerances are chosen as 0.0 eV and $0.01\ eV/\text{Å}$, respectively. These parameters are selected to optimize the convergence of the calculations[15].

### Structure representation and neural networks

The on-lattice representation coverts the atomic structure into a digit matrix, which will be deciphered by neural networks. The conversion is done through a voxel grid that separates the 3D material model into uniform cubes. Each grid acquires a digit value (voxel) according to its enclosed atom type or vacancy. For bcc structure, the largest grid we can use, which can fully distinct all lattices and yield the smallest voxel grid dimensionality, is $a/2$, where $a$ is the lattice constant of the crystal (see Supplementary Note 1).

The neural network, taking the representative structure and chemistry digits (neurons) as input, process them through the hidden layers, outputting the energy barriers. The connections between neurons in the hidden layers imitate the physical interactions between atoms and atom-vacancy. Representing the interaction strength (contribution to the migration barrier), the weights associated with the connections are adjusted during training. To understand the influence of network architecture on prediction performance, we train a series of neural networks with varying number of layers and number of neurons in each layer (Supplementary Note 3). As the number of neurons in each hidden layer increases from 16, 32, 64, to 256, the testing MAE rapidly decreases, followed by convergence at 128 that is enough to explicitly describe all the local neighbors of a vacancy (Supplementary Fig. S14). By testing the different number of layers, the final network structure with 4 hidden layers and 128 neurons in each layer was selected for simulating the diffusion in the equimolar NbMoTa alloy, owing to its robustness in concentrated solid solutions. In addition, we separately train a convolutional neural network (CNN) to compare with the simple neural network. The CNN comprises four convolutional layers that compress the 3D neuroma map to $1 \times 128$ dimension for barrier prediction. The architecture of CNN is depicted in Supplementary Fig. S17 and described in Supplementary Note 3. Likely resulting from adaptive learning spatial hierarchies of features from input 3D atomic structure, CNN exhibits slightly enhanced predictive performance (Supplementary Fig. S20).

The training data are generated from 46 different compositions, which uniformly sample the Nb−Mo−Ta diagram (Supplementary Fig. S18 and Supplementary Table S3). In Supplementary Note 4, we carefully study and discuss the number of compositions required to train a highly accurate network for predicting the complete ternary space. Each composition model contains 2000 atoms, giving rise to 16,000 diffusion barriers. The total 736,000 data points are split into training dataset (95% of total data) and validation dataset (5%). All the compositions and their data points are summarized in Supplementary Table S3. After validation, the neural network is tested for barrier

prediction in unseen compositions (which are not used for training or validation) and in atomic configuration with different sizes. For example, Supplementary Fig. S4 shows the testing results for the new compositions, $Nb_{10}Mo_{10}Ta_{80}$, $Nb_{20}Mo_{60}Ta_{20}$, $Nb_{40}Mo_{30}Ta_{30}$, and the average MAE is around 0.018 eV. Notably, the neural network preserves the consistently high accuracy for different-sized systems containing 512, 2000, and 6750 atoms, indicating scalability.

## Neuron kinetics

The neuron map enables efficient modeling of vacancy kinetics through the exchange of neurons, referred to as neuron kinetics. By converting the atomic configuration into a neuron map just once, the neural network simulates vacancy jumps simply by swapping two neurons within the map (vacancy and one of its nearest-neighboring neurons). This streamlined process allows to efficiently model tens of millions of vacancy jumps. Importantly, it is worth noting that each jump iteration involves the exchange of only two neurons, as depicted in Fig. 1b.

Vacancy jump is carried on the neuron map based on the kinetic Monte Carlo (kMC) algorithm. Diffusion occurs through vacancy (vacancy neuron) jump to its nearest-neighboring sites, and each site has a jump rate defined by $k_i = k_0 \exp(-E_i/k_B T)$, where $E_i$ is the energy barrier along jump path $i$, $k_B$ is Boltzmann constant, $T$ is temperature, and $k_0$ is an attempt frequency. The vacancy diffusion barriers associated with the eight jump paths are obtained from the neural network. The total jump rate for the current vacancy configuration is $R = \sum_{i=1}^{8} k_i$, i.e., the sum of all individual elementary rate. To simulate kinetic evolution, we first draw a uniform random number $u \in (0,1]$ and select a diffusion path, $p$, which satisfies the condition[46], $\sum_{i=1}^{p-1} k_i/R \le u \le \sum_{i=1}^{p} k_i/R$. The vacancy jump along path $p$ is then executed by exchanging the vacancy with the selected neighboring neuron (neuron digit swapping), resulting in an updated neuron map for the next iteration.

## Static Monte Carlo and molecular dynamics simulation

We perform static Monte Carlo (MC) simulations coupled with molecular dynamics to reveal the chemical order determined by enthalpy (mainly thermodynamics). In each MC trial, a pair of atoms is randomly selected for type swap. The acceptance probability is according to the $\exp(-\Delta H/k_B T)$ in Metropolis algorithm[47]. The term $\Delta H$ is the enthalpy change after swap, therefore, the chemical evolution and ordering is predominately contorted by enthalpy. The MC swaps are followed by MD equilibration. For the systems consisting 1024 atoms, we perform 18,000 swap attempts (each atom on average subjected to 18 swaps) and 600 ps MD equilibrium. Supplementary Fig. S9 shows the local order as a function of MC step for temperatures from 100 to 3000 K. To study B2 cluster growth, we perform the MC and MD simulation in a large model (128,000 atoms). There are totally 135,000 swaps coupled with 150 ps MD equilibrium. Unlike diffusion-mediated B2 cluster growth, the clusters grow in a uniform and homogeneous manner (Supplementary Fig. S10).

## Local chemical order parameter

To quantify the degree of chemical order, we use the non-proportional parameter[48] $\delta_{ij} = N_{ij} - N_{0,ij}$, where $N_{ij}$ denotes the actual number of pairs between atoms $i$ and $j$ in the first nearest-neighboring shell, and $N_{0,ij}$ represents the average number of pairs in random solutions. A positive $\delta_{ij}$ means a favored and increased number of $i$-$j$ pairs, indicating element $i$ tends to bond with element $j$. A negative $\delta_{ij}$ indicates unfavored pair, meaning $i$ and $j$ repel each other. Random solid solution has $\delta_{ij} = 0$.

## B2 cluster analysis

Mo and Ta tend to attract each other and form the B2 structure. The B2 unit cell has a simple bcc structure and comprises two species, Ta and Mo, orderly located in the cube corners or center. The unit cell can have either Ta or Mo-centered pattern. Because of the high concentration of Nb in the equimolar NbMoTa alloy, we characterize a unit as B2 when 3/4 of the Ta nearest neighbors are Mo, or 3/4 of the Mo nearest neighbors are Ta. To analyze the B2 cluster, the identified individual B2 units are gathered into individual group according to distance criterion. Two B2 units can have volume-, face-, edge-, and point-sharing at distance $\sqrt{3}a/2$, $a$, $\sqrt{2}a$, $\sqrt{3}a$ (i.e., 5th shell), respectively, where $a$ is lattice constant (illustrated in Supplementary Fig. S7). Choosing the cutoff distance as half of the 5th shell and 6th shell, the spatial distribution and size of all B2 clusters can be successfully characterized. During the kinetic annealing, clusters can be reduced or annihilated, which causes clusters appearance or disappearance from time to time. The fluctuation hinders visualization and analysis of stable B2 cluster evolution. To address this issue, we search and identify the persist clusters that exist all the time during annealing. Focusing on the persistent cluster provides a clear evolution of cluster growth (Fig. 5).

## Data availability

The diffusion data in this study have been deposited in the Zenodo under accession code https://doi.org/10.5281/zenodo.7714650.

## Code availability

All source codes of NNK are available at the GitHub repository https://github.com/UCICaoLab/NKK[49].

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

## Acknowledgements

This research was primarily supported by the National Science Foundation Materials Research Science and Engineering Center program through the UC Irvine Center for Complex and Active Materials (DMR-2011967). P.C. acknowledges the support from the U.S. Department of Energy, Office of Basic Energy Sciences, under award DE-SC0022295, in the work on SRO and defect interaction. The authors thank L. Qi of UMich for valuable discussions regarding cluster expansion.

## Author contributions

P.C. conceived the research idea, wrote the manuscript, and produced the figures with inputs from B.X. B.X. developed the model, implemented the code, and performed simulation and modeling. T.J.R. and X.P. reviewed and edited the manuscript. All authors contributed to data analysis and project discussion.

## Competing interests

The authors declare no competing interests.
