## [Peer Review File · Nature Communications]

Neural Network Kinetics for Exploring Diffusion Multiplicity and Chemical Ordering in Compositionally Complex MaterialsReviewers' comments:

Reviewer #1 (Remarks to the Author):

The manuscript discusses diffusion involving atom transport and its significance in various processes such as precipitation and phase nucleation. The paper introduces a “neural network kinetics (NNK) scheme” for simulating diffusion-induced chemical and structural evolution in complex chemical environments. The framework utilizes on-lattice structure, chemistry representation, and artificial neural networks. While this work is timely, several clarifications are necessary:

1. In Figure S1, the authors have mentioned about the digital matrix, vacancy, and the nearest neighbor atoms. However, it is not clear how these nearest neighbor atoms are found? Is the distance between them calculated using Euclidean or Manhattan or other distance formula?
2. On page 4, line 92, the authors have mentioned that the representation is rotational non-invariant. The representation is a neural map which is a digital matrix. Please clarify.
3. In Section 3 of the supplement, the authors note that they calculated the activation barrier in every direction, leading to a total of 64,000 barriers. How was this determined?
4. In Section 3 of the supplement, the authors state that they divided the dataset into 80% for training and 20% for testing. It is necessary to explore alternative splits and utilize cross-validation to ensure the model generalizes effectively and is not biased by the chosen datasets.
5. In Section 3 of the supplement, the authors indicate that they trained a neural network with 4 hidden layers and 128 hidden neurons for every cut-off distance. Why are 4 hidden layers deemed optimal? Does it not result in overfitting?
6. In Supplementary Section 4, the authors stated that the dataset is split into 95% for training and 5% for validation. However, in Supplementary Section 3, they indicate a split of 80% and 20%. The authors should clarify the discrepancy between these two sections regarding the dataset splits.
7. In Supplementary Section 4, the authors note their experimentation with convolutional neural networks comprising four convolutional layers and a single output layer. Were the CNNs tested with a varying number of convolutional layers. Additionally, did the authors explore training a model that combines CNN layers with traditional neural network layers? This step is essential to ascribe layer independence to the predictions.
8. In Supplementary Section 4, when discussing the CNN model, the authors indicate the use of ReLU as an activation function. Please explain the choice of ReLU over other potential activation functions.
9. In Figure 2a, how are atoms assigned to specific layers? Moreover, there are zeroes present in the atoms, neurons, and vacancy layers. How does the model differentiate among these zeroes?

10. On page 7, line 189, the authors present a formula for determining the probability of each possibility. What is the genesis and rationale for this formulation?

11. On page 13, line 360, the authors used 95% of the data for training their model and the remaining 5% for validation. The split seems extreme, and varied training/test data set splits are suggested to eliminate any noise/bias.

12. On page 14, line 359, the authors state there are 16,000 diffusion barriers. However, in Supplementary Section 3, they reference 64,000 diffusion barriers. The authors should address and clarify this discrepancy.

13. On page 14, line 360, the authors note a total of 736,000 data points. How do the authors compute this value?

Reviewer #2 (Remarks to the Author):

Major Comments

Overall this is some interesting work but does yet to be improved to meet the high standard of nature communications. 3D convolutional networks are very well-established in machine learning, although their application to atomistic simulations is interesting.

More work needs to be done in order to establish this technique as superior to existing methods, namely cluster expansion (CE):

A) They should better-clarify the advantages of their method over cluster expansion, which is mentioned only very briefly

Relative to CE:

- 1) Is this method more accurate
- 2) Is it easier to generalize? Does it require less training points?
- 3) Is it faster to run?
- 4) Is it simpler to use?

Or any other advantage, this must be made clearer in the text. [Ideally with benchmarks, time permitting, else with greater speculative arguments]

B) They could establish more rigorously the methods' ability to scale to a high number of elements by using some of the many new machine learning potentials (MLPs) that can work on a larger number of elements. As of right now they only test up to 3 elements which has been long-established using cluster expansion(<https://www.sciencedirect.com/science/article/pii/S0927025607000158>)

There is probably some benchmark that can be done using more elements. See <https://github.com/ACEsuit/mace#materials-project> for an example

C) In theory by expanding the conv net and including interstitial points the model has the potential to cover off-lattice transformations, could this be tested?

C) The citations only seem to cover almost entirely famous high-impact publications but it seems like the authors are not fully caught up in the literature in techniques of mapping atomic configurations to properties

Minor Comments

D) They say that atomic potentials can't be 'directly' trained on diffusion barriers but this seems confusing and misleading. They are trained on the relevant input structures and compute the energy difference in the same way DFT would. Linear regression could be 'trained directly' but that is somewhat meaningless to mention.

They should clarify the advantages of this 'direct' training

E) the use of jet colour scheme should be avoided as it give a false impression of gradients, the viridis colorscheme is more accurate to the eye [and is more accessible in black and white or for colorblind readers]

G) There is a growing body of literature using 3D conv nets, e.g., in MRI medical imaging, it would be nice if the authors made at least some small effort to establish what is occurring in the field.

H) The introduction says that current descriptors scale quadratically with the number of elements, they should look at recent papers by the Ceriotti [<https://journals.aps.org/prmaterials/abstract/10.1103/PhysRevMaterials.7.045802>] and Csányi [<https://journals.aps.org/prl/abstract/10.1103/PhysRevLett.131.028001>] who have made great progress in reducing the scaling with number of elements.

Reviewer #3 (Remarks to the Author):

This NNK framework will be an excellent tool in the area of CCAs to explore properties for vast compositional spaces. This manuscript should be published before fixing some issues as illustrated below.

1. In the manuscript it is clearly mentioned how to generate the neuron map. However, why not use a graph neural network directly instead of creating the neuron map? Please add justifications for creating neuron maps over graph neural network-based approaches.
2. The link under Data Availability is not the correct link.

Reply to reviewers:

The comments from three reviewers are presented in black, with our responses highlighted in blue. Revisions in the Manuscript and Supplementary Information (SI) are highlighted by red color, and a summary of the changes is provided at the end of this document.

Reviewer #1:

The manuscript discusses diffusion involving atom transport and its significance in various processes such as precipitation and phase nucleation. The paper introduces a “neural network kinetics (NNK) scheme” for simulating diffusion-induced chemical and structural evolution in complex chemical environments. The framework utilizes on-lattice structure, chemistry representation, and artificial neural networks. While this work is timely, several clarifications are necessary:

Reply: We thank the reviewer for the summary of the work, and for the comments of “the introduced neural network kinetics scheme for addressing diffusion in complex chemical environments is timely”. Below, we are providing our responses to the raised questions.

1. In Figure S1, the authors have mentioned about the digital matrix, vacancy, and the nearest neighbor atoms. However, it is not clear how these nearest neighbor atoms are found? Is the distance between them calculated using Euclidean or Manhattan or other distance formula?

Reply: In Figure S1(h-j), we illustrate a vacancy (vacant lattice site) and its nearest neighboring atoms. To determine the nearest neighbors of lattice site in crystal structure, the Euclidean distances between the reference site and all its neighbors are first measured. For body-centered cubic (BCC) structure, the first and second nearest neighboring distances are $\sqrt{3}a/2$ and a , respectively, where a is the lattice constant. Using a cutoff distance of $0.5 * (\sqrt{3}a/2 + a)$, one can identify all 1st nearest neighbors. By varying the cutoff distance, different nearest neighbors can be classified. We have revised the Figure S1 caption to clarify this.

2. On page 4, line 92, the authors have mentioned that the representation is rotational non-invariant. The representation is a neural map which is a digital matrix. Please clarify.

Reply: We thank the reviewer for the comment. The rotational non-invariance of neural map (digital matrix) is critical for predicting path-dependent barriers of vacancy diffusion. For a given atomic configuration that includes a vacancy, there are eight migration paths associated with the vacancy (corresponding to the eight 1st nearest neighbors). The key challenge lies in how to predict these distinct migration barriers from one neural map (atomic configuration). To address this, we introduce a ‘reference direction’, which aims to mark the diffusion path of interest. By performing rotation and mirroring operations on the atomic configuration, we can align the diffusion direction of interest with this reference. Hence, unique digital matrices and digital vectors can be generated for each individual diffusion path, preserving structural symmetry. Figure R1 below exemplifies

this process, showing how diffusion paths 2 and 3 are aligned with the reference direction (indicated by red arrow). Figure 2 of our manuscript and Figure S2 of the SI details these operations and the resultant matrices, which are linked to the eight vacancy diffusion paths.

Figure R1. Aligning diffusion paths 2 and 3 with the reference direction through rotation. It produces two digital matrices and vectors corresponding to the two paths.

This rotational non-invariant feature of digital matrix can also be understood from the handwritten digit recognition. For instance, when the MNIST database's handwritten '6' is rotated by 180 degrees, it resembles a '9', as shown in Figure R2. Despite the pixel values in the matrices being unchanged, the orientation relative to the reference direction (denoted by the arrow) allows for the correct interpretation. To make it more clear for future readers, we have expanded the discussion on the atomic representation and its non-invariance in the revised SI.

Figure R2. Rotational non-invariance for handwritten digit recognition.

3. In Section 3 of the supplement, the authors note that they calculated the activation barrier in every direction, leading to a total of 64,000 barriers. How was this determined?

Reply: Section 3 of the SI aims to explore how a cutoff distance impacts the predictive performance of our machine learning models for vacancy migration barriers. To do so, we select four alloy compositions: equimolar NbMoTa, Nb₅₀Mo₂₅Ta₂₅, Nb₂₅Mo₅₀Ta₂₅ and Nb₂₅Mo₂₅Ta₅₀. For each composition, we simulate atomic configurations comprising 2,000 atoms (i.e., lattice sites). Considering that each vacancy can migrate in one of eight possible directions, this results in 16,000 unique migration barriers per composition (2,000 vacancies × 8 directions). Consequently, by studying four distinct compositions, we determine a total of 64,000 barriers (16,000 barriers per composition × 4 compositions).

To accurately calculate these barriers, we employ the Nudged Elastic Band (NEB) method, which identifies the saddle point between the initial and final energy minimum configurations. The details regarding the parameters used in our NEB calculations are outlined in the Methods section under “Material System and Diffusion Barrier Calculation”.

4. In Section 3 of the supplement, the authors state that they divided the dataset into 80% for training and 20% for testing. It is necessary to explore alternative splits and utilize cross-validation to ensure the model generalizes effectively and is not biased by the chosen datasets.

Reply: Following the reviewer’s suggestion, we performed 5-fold cross-validation, with the results shown in the Figure R3. Specifically, the dataset is divided into five folds to train five separate models. The small error bars of the MAE (mean absolute error) summarize the predictive error of the five models, underscoring the generalizability. We have incorporated these new cross-validation results into the revised SI.

It’s worth noting that our primary objective of Section 3 is to understand the influence of cutoff distance on vacancy migration barrier. As the cutoff distance increases, the MAE rapidly decreases, converging near a cutoff distance of 7.5 Å (corresponding to the 8th nearest neighboring shell). This suggests that any further increase in the cutoff distance would contribute negligibly to the neural network model prediction, implying that vacancy barrier is predominately influenced by the atoms within the 8th shell. With the obtained cutoff, we develop refined neural network models that encompasses the entire compositional space.

Figure R3. Five-fold cross-validation of neural network for different cutoff distances. The red and blue curves represent the training and validation mean absolute error (MAE), respectively. The error bars represent the standard deviations of model prediction errors.

5. In Section 3 of the supplement, the authors indicate that they trained a neural network with 4 hidden layers and 128 hidden neurons for every cut-off distance. Why are 4 hidden layers deemed optimal? Does it not result in overfitting?

Reply: The number of hidden layers and the number of neurons in each layer are the two critical parameters of neural network architecture. To understand how the architecture influences the performance and ultimately help us to determine the model architecture, we train separate neural networks, with different numbers of hidden layers (from 1-4) and number of layer neurons (16 – 256). Figure R4a shows the prediction MAEs of these models (also shown in Figure S15 of SI). We found that the model performance converges when the number of hidden layers and number of hidden neurons reaches 4 and 128, respectively. Physically, the convergence on 128 neurons has physical meaning as they can explicitly capture all 112 input neurons (i.e., the total number of neighboring atoms of a vacancy up to the 8th shell).

Figure R4. (a) Validation errors of neural network models with varying number of hidden layers and neurons. (b) Training errors (shown in red) and validation errors (in blue) across five-fold cross-validation for different numbers of hidden layers.

To understand the influence of hidden layers, we separately train neural networks with varying number of layers, from 1 to 4. Figure R4b shows both training and validation errors at varying number of hidden layers from five-fold cross-validations. The result indicates that the neural network model reaches a performance convergence as the number of hidden layers increases to 4.

6. In Supplementary Section 4, the authors stated that the dataset is split into 95% for training and 5% for validation. However, in Supplementary Section 3, they indicate a split of 80% and 20%. The authors should clarify the discrepancy between these two sections regarding the dataset splits.

Reply: Section 3 and Section 4 in SI are to address different aspects of developing neural network models for predicting diffusion in the entire space. We are providing the detailed interpretation.

In Section 3, we want to reveal the impact of cutoff distance on vacancy migration barrier (as described in response to the Questions 3 above). To do so, we compute and prepared 64,000 barriers from four alloy compositions (16,000 barriers per composition \times 4 compositions). Considering the size of the dataset, we adopt the classic 80/20 splitting to ensure enough data to be used for validation. As the cutoff distance increases, the MAE rapidly decreases, converging near a cutoff distance of 7.5 Å (corresponding to 8th nearest neighboring shell). This suggests that any further increase in the cutoff distance would contribute negligibly to the neural network model prediction, implying that vacancy barrier is predominately influenced by the atoms within the 8th shell.

After determining the cutoff distance, we develop two versions of machine learning models, neural network and convolutional neural network, for learning and predicting vacancy diffusion barrier in the entire composition space of Nb-Mo-Ta. For this purpose, we compute and generate 736,000 vacancy barriers from 46 compositions (16,000 barriers from each composition). These 46 compositions have been shown in the ternary phase diagram in Figure S12 of SI. Considering the large dataset size, the 95/5 splitting allows model training and generalization, meanwhile giving a large dataset ($736,000 \times 5\% = 36,800$) for validation.

It is worth noting that **the testing performance of the trained neural networks is evaluated using newly generated data from other (unseen) compositions, which are not used for training and validation.** Figure R5 shows the testing results for these new compositions, $\text{Nb}_{10}\text{Mo}_{10}\text{Ta}_{80}$, $\text{Nb}_{20}\text{Mo}_{60}\text{Ta}_{20}$, $\text{Nb}_{40}\text{Mo}_{30}\text{Ta}_{30}$, and the average MAE is around 0.018 eV, implying the **generalizability**. Notably the neural network preserves the consistent high accuracy for different sized systems containing 512, 2000, and 6750 atoms, indicating **scalability**.

To avoid possible future confusion, we prepare tables summarizing all the datasets used to train all the neural network models and have included them in the revised SI.

Figure R5. Performance evaluation of neural network in predicting diffusion barriers in unseen compositions and varying system sizes (scalability). Three compositions, including $\text{Nb}_{10}\text{Mo}_{10}\text{Ta}_{80}$, $\text{Nb}_{20}\text{Mo}_{60}\text{Ta}_{20}$, $\text{Nb}_{40}\text{Mo}_{30}\text{Ta}_{30}$, and three systems containing 512, 2000, and 6750 atoms are shown.

7. In Supplementary Section 4, the authors note their experimentation with convolutional neural networks comprising four convolutional layers and a single output layer. Were the CNNs tested with a varying number of convolutional layers. Additionally, did the authors explore training a model that combines CNN layers with traditional neural network layers? This step is essential to ascribe layer independence to the predictions.

Reply: The dimensionality of input images to convolutional neural networks is $9 \times 9 \times 9$. In each convolutional layer, the filter size is 3. Thus, the spatial dimension gradually decreases from $9 \times 9 \times 9 \times 1$ to $1 \times 1 \times 1 \times 128$ after four convolutional layers. This is why the model contains four convolutional layers. Figure R6a illustrates this convolutional neural network architecture, consisting of one input layer, four convolutional layers, and one output layer.

To answer the reviewer's question about the impact of traditional neural network layers on prediction performance, we add two additional neural network layers to the CNN model, and the model architecture is shown in Figure R6b. Utilizing the same dataset, we assessed the model's performance both with and without these traditional neural network layers. Figure R7 shows the predictions of the two models on diffusion barriers in three unseen compositions, namely

$\text{Nb}_{10}\text{Mo}_{10}\text{Ta}_{80}$, $\text{Nb}_{20}\text{Mo}_{60}\text{Ta}_{20}$, and $\text{Nb}_{40}\text{Mo}_{30}\text{Ta}_{30}$. The CNN model alone yielded prediction errors of 0.013, 0.018, and 0.021 eV for these systems, respectively. When we incorporated the two additional neural network layers, the errors remained similar: 0.014, 0.018, and 0.021 eV. These findings suggest that the inclusion of traditional neural network layers does not significantly increase the CNN model's prediction accuracy. This implies that the CNN model captures the pattern and sequence of the neural map (atomic configuration), accurately linking it to the diffusion barriers.

Figure R6. (a) Architecture of the original CNN model, consisting of one input layer, four convolutional layers. (b) Revised CNN model, with two traditional neural network layers being incorporated.

Figure R7. Comparative analysis of performance in predicting diffusion barriers for three unseen compositions. (a) Performance of the original CNN model. (b) Performance of the revised CNN model, with two additional neural network layers added.

8. In Supplementary Section 4, when discussing the CNN model, the authors indicate the use of ReLU as an activation function. Please explain the choice of ReLU over other potential activation functions.

Reply: We selected ReLU as our activation function over others like Sigmoid or Tanh due to its advantageous of non-linearity and computational efficiency. Physically, ReLU linearizes positive inputs while nullifying negative ones, effectively capturing the non-linear interactions between atoms as represented by neurons. Additionally, ReLU does not involve expensive calculations like exponentials or division, making the process more computationally efficient.

9. In Figure 2a, how are atoms assigned to specific layers? Moreover, there are zeroes present in the atoms, neurons, and vacancy layers. How does the model differentiate among these zeroes?

Reply: To digitally represent atomic structures, we employ an on-lattice approach, transforming them into a digital matrix. This process begins by partitioning the crystal structure into a grid of voxels, ensuring that each lattice site is centered within a voxel. Subsequently, we allocate different atom types to their respective voxels, encoding the voxel's value based on the type of atom it

contains. Voxels without any atoms are encoded as zero. Through this method, atoms are accurately positioned in specific layers of voxels, reflecting their actual locations within the crystal structure. Section 2 of the SI discusses this on-lattice methodology and the representation of chemical structures, including details on determining pixel size and the conversion process from atomic structure/type to voxel values.

The neural network model discerns the overall sequence and pattern in the digital matrix, not individual zeros. In a perfect crystal structure (BCC here), the digital matrix displays a consistent sequence of non-zero and zero digits. However, the introduction of a vacancy alters this structure by adding an additional zero at the corresponding location. This alteration in the digital sequence is what the neural network is trained to detect and learn from, enabling it to predict associated properties.

10. On page 7, line 189, the authors present a formula for determining the probability of each possibility. What is the genesis and rationale for this formulation?

Reply: To quantify the jump probability for each of the eight possible paths, we define a parameter $p_i = \exp(-E_i/k_B T) / \sum_{j=1}^8 \exp(E_j/k_B T)$. It is known that, for each jump path, the vacancy jump rate is $k_i = k_0 \exp(-E_i/k_B T)$, where E_i is the energy barrier along path i , k_B is Boltzmann constant, T is temperature, and k_0 is an attempt frequency. The total jump rate for a vacancy is the sum of all elementary rates, i.e., $R = \sum_{j=1}^8 k_j = \sum_{j=1}^8 k_0 \exp(-E_j/k_B T)$. The probability of diffusion path i being selected is the ratio between k_i and $\sum_{j=1}^8 k_j$. Thus, the probability equals to $p_i = k_i/R = k_0 \exp(-E_i/k_B T) / \sum_{j=1}^8 k_0 \exp(-E_j/k_B T) = \exp(-E_i/k_B T) / \sum_{j=1}^8 \exp(E_j/k_B T)$. Figure 4a of the manuscript shows the physical meaning of p_i for the two limiting cases: directional jump and random jump.

11. On page 13, line 360, the authors used 95% of the data for training their model and the remaining 5% for validation. The split seems extreme, and varied training/test data set splits are suggested to eliminate any noise/bias.

Reply: We believe the reviewer raised similar point in the question 6. To develop a machine learning models predicting vacancy diffusion barrier in the entire composition space of Nb-Mo-Ta, we compute and generate 736,000 vacancy barriers from 46 compositions (16,000 barriers from each composition). These 46 compositions have been shown in the ternary phase diagram in Figure S12 of SI. Considering the large dataset size, the 95/5 splitting allows model training and generalization, meanwhile giving large dataset ($736,000 \times 5\% = 36,800$) for validation. The testing performance of the trained neural networks is evaluated using newly generated data from other (unseen) compositions, which are not used for training and validation. Figure R5 has shown the testing results for these new compositions, $\text{Nb}_{10}\text{Mo}_{10}\text{Ta}_{80}$, $\text{Nb}_{20}\text{Mo}_{60}\text{Ta}_{20}$, $\text{Nb}_{40}\text{Mo}_{30}\text{Ta}_{30}$, and the average MAE is around 0.018 eV, implying the generalizability.

12. On page 14, line 359, the authors state there are 16,000 diffusion barriers. However, in

Supplementary Section 3, they reference 64,000 diffusion barriers. The authors should address and clarify this discrepancy.

Reply: The 16,000 diffusion barriers are obtained from one alloy composition. In Supplementary Section 3, we consider four compositions, which give rises to a title of 64,000 barriers (16,000×4). We have provided a table summarizing all the dataset in the amended SI.

13. On page 14, line 360, the authors note a total of 736,000 data points. How do the authors compute this value?

Reply: We uniformly sample 46 compositions in the ternary space. For each composition, we use NEB method to compute 16,000 energy barriers, and the 46 compositions result in a total of 736,000 data points. We have induced a newly prepared table in SI to show all the compositions and the data points generated.

Reviewer #2:

Major Comments

Overall this is some interesting work but does yet to be improved to meet the high standard of nature communications. 3D convolutional networks are very well-established in machine learning, although their application to atomistic simulations is interesting.

Reply: We thank the reviewer for finding the work interesting. The primary objective of the study is to develop a computational scheme that can tackle the challenges for modeling atomic diffusion and the resulting formation of chemically ordered structures in compositionally complex materials. The neural network, as a part of the scheme, is tailored to learn and predict vacancy diffusion in concentrated chemical environments and vast compositional space. We consider, however, the following three concepts as pivotal features introduced by this work, targeting the emerging challenges in compositionally complex materials:

(i) Neuron map (on-lattice) representation. This interpretable on-lattice representation, which fully captures the chemistry and structure of crystals yields an ultra-small feature size $O(n)$.

(ii) Rotational non-invariance. The rotational non-invariance of the neuron map enables prediction of path-dependent diffusion barriers (vector values) from a single neuron map, and hence fully replicates atomic diffusion jumps and its induced structure evolution.

(iii) Scalability. The framework, trained by small models, can be applied directly to investigate diffusion in large systems without sacrificing accuracy.

Using the framework, we study temperature-dependent local chemical ordering in a refractory NbMoTa alloy and reveal a critical temperature at which the B2 order reaches a maximum. This

critical temperature is interpreted by microscopic diffusion theory and our proposed jump randomness map.

Detailed replies to the reviewer's major and minor comments are included below.

More work needs to be done in order to establish this technique as superior to existing methods, namely cluster expansion (CE):

A) They should better-clarify the advantages of their method over cluster expansion, which is mentioned only very briefly

Reply: Thank the reviewer for suggesting us to compare our current method with cluster expansion (CE). We understand that the CE was designed to predict **thermodynamic properties** of multicomponent systems. In this computational approach, the total energy of a configuration is computed by summing up a series of clusters (for instance, single, pair, triplet, and large group of atoms). For **kinetics** problems such as vacancy diffusion, the governing property is the underlying diffusion activation energy ΔE , i.e., the energy difference (barrier) between transition state and the initial energy minimum.

In Figure R8, we schematically illustrate vacancy diffusion and its corresponding potential energy landscape. The process starts from an initial state E_i , through a transition state E_s , and leads to the neighboring local minimum, i.e., final state E_f . The CE has been adopted to predict the energies of local minimum states. However, addressing the transition state presents specific challenges in CE method. We will address these in detail later in response to the reviewer's questions about comparison with CE.

In the classic application of CE for predicting configurational energy, the initial critical steps involve designing unique clusters based on lattice symmetry and choosing an optimal cluster set, which can be a time-consuming process. For example, the diffusion of a vacancy can be influenced by the atoms in its 8th neighboring shell. It has been noted that it can take a number of weeks to months to select optimal clusters when the 8th nearest neighboring atoms is considered (1).

Figure R8. Schematic illustration of vacancy diffusion and the corresponding diffusion energy landscape. (a) Vacancy diffusion states from an initial state, through saddle point, and leads to the final state. (b) The energy barrier ΔE , i.e., the energy difference between transition state and the initial energy minimum, is the governing value for diffusion. The key task is to accurately and efficiently predict these barriers in compositionally complex materials.

Relative to CE: 1) Is this method more accurate, 2) Is it easier to generalize? Does it require less training points? 3) Is it faster to run? 4) Is it simpler to use? Or any other advantage, this must be made clearer in the text.

[Ideally with benchmarks, time permitting, else with greater speculative arguments]

Reply: In the literature, there have been attempts to use CE method for predicting saddle point energy in binary alloys. To the best of our knowledge, there is only one study (2) focusing on predicting vacancy barriers in compositionally complex alloys (ternary alloys). In the following, we will delve into the challenges associated with employing CE for this task and assess its performance in these contexts.

(i) Saddle point representation. To model the transition state (saddle point) for energy prediction using CE, the strategy is to introduce artificial atoms, as shown in Figure R9. In a binary system, the jumping species could be either atom type 1 or 2. To differentiate, two additional species, type

3 and type 4, are introduced. For example, the top panel of the figure shows how the transition state involving atom 1 jump (black donut) jump is represented. This increases chemical complexity and the intricacy of cluster design. In a N-component system, N extra species need to be defined, leading to a total of $2N+1$ species (including vacancy).

Figure R9. Transition state representation in CE. (left) The two transition states involve different jumping atom types, 1 and 2. (right). The new species, types 3 and 3, are introduced to model the two transition states. Figure is cited from ref. 1.

(ii) CE prediction performance in ternary alloy. To predict vacancy diffusion barrier in ternary system (Al-Mg-Zn), the CE is combined with a quartic function of the reaction coordinate (2). Figure R10 compares the predicted diffusion barriers with the ground truth obtained from NEB calculation. The dataset includes 500 diffusion barriers. The mean error is 0.0451 eV, approximately 10% of the average actual barrier of around 0.5 eV. This is contrasted with our model, as shown in Figure 2 of the manuscript, which achieves a significantly lower mean error of 1.2%, an order of magnitude lower than that of CE. We discussed this with the authors of the work, Prof. Liang Qi from University of Michigan, and learned that they encountered the same above described challenges in using cluster representation for saddle point configurations.

Figure R10. The performance of CE in predicting diffusion barrier in ternary alloy, Al-Mg-Zn. Comparison between the predicted barrier and the NEB calculation shows a mean error of 0.0451 eV. The data is cited from ref. 2.

(iii) Our atomic neural network kinetics model. We highlight the four key features associated with our model, which render its highly accurate barrier prediction. It is noted that the model performance in highly accurate barrier prediction is not just for a single alloy composition but across a wide range of varying alloy compositions (the entire compositional space of the ternary alloys).

a. Neuron map representation of atomic structure and chemistry: The neuron map (on-lattice) representation precisely captures atomic structure and composition. Its dimension $O(N)$ scales linearly with the number of atoms N , and has the lowest dimensionality possible as a crystal descriptor. Critically, determining the neuron map is simple and involves simple calculation (avoiding the painstaking parameter tuning in other methods, such as cluster design in CE).

b. Predict performance: The model exhibits high accuracy in barrier prediction. For instance, the mean absolute errors (MAE) for dilute solution $\text{Nb}_{90}\text{Mo}_5\text{Ta}_5$ and concentrated solution $\text{Nb}_{33}\text{Mo}_{33}\text{Ta}_{33}$ are 0.011 and 0.021 eV, respectively. The error is smaller than 1.2% of the true diffusion barrier (Figure 2b and Figure S17).

c. Generalization and predicting in entire compositional space: More importantly for compositionally complex materials possessing a vast compositional space, the current method, trained on dozens of compositions, shows remarkable predictability for new (previously unseen) compositions, allowing accurate mapping of the entire ternary space (Figure 2c).

d. Scalability and efficiency: Our model demonstrates scalability with system size. This size scalability is shown by accurate barrier predictions in larger NbMoTa systems. For example, the neural network preserves a consistent high accuracy for different sized systems containing 512, 2000, and 6750 atoms.

e. High efficiency in modeling diffusion: neural map originates from its simplicity to mirror vacancy jumps through the swapping of neurons (digits). With only one-time conversion of atomic configuration to neuron map, vacancy jumps and chemical evolution can be simulated by swapping two digits of neural map. In this way, millions of vacancy jumps can be modeled efficiently, with each jump iteration involving the action of just two neurons (illustrated in Figure R15b). Using one single CPU, the model evolves 10 million diffusion jumps in a large system containing 128,000 atoms within two days.

Compared with CE, our introduced neuron map representation and the computational scheme are more accurate (low error prediction), earlier to generalize (will also demonstrate below about the more complex system), and capable of predicting vacancy barriers in the entire space using small training data (Figure 2c of the manuscript), and easy and fast (Figure 5, evolving 10 million jumps in large system). We thank the reviewer very much for making these suggestions and comments. We have included the above discussion and made them clear in the revised text of manuscript and new section 6 of SI.

B) They could establish more rigorously the methods' ability to scale to a high number of elements by using some of the many new machine learning potentials (MLPs) that can work on a larger number of elements. As of right now they only test up to 3 elements which has been long-established using cluster expansion

(<https://www.sciencedirect.com/science/article/pii/S0927025607000158>).

There is probably some benchmark that can be done using more elements. See <https://github.com/ACEsuit/mace#materials-project> for an example

Reply: We studied the paper recommended by the reviewer and found that it applied the CE to predict the configurational energy of the Al-Mg-Si ternary system (like what is cited in references 8 and 9 of our manuscript). In the study, the initial and final configurational energies, predicted by CE, were used in conjunction with the static Metropolis algorithm to model trial changes and the evolution of the material's chemistry. However, it's important to highlight that this approach does not explicitly account for vacancy jump kinetics, which necessitates path-dependent diffusion barriers—a significant challenge that our study addresses.

Our main focus of the current study is to introduce a neural network kinetics scheme (i) for predicting path-dependent migration barriers and individual atom jumps in compositionally complex materials, and (ii) using the ternary Nb-Mo-Ta alloy as an example, it uncovers the temperature-dependent local chemical ordering kinetics and a critical temperature at which the B2 order peaks. Addressing the reviewer's question about testing with additional elements, we present preliminary results on the model's scalability to more complex systems, including quaternary and quinary alloys.

Using DFT (density functional theory) and NEB calculations, we calculated 20,480 vacancy diffusion barriers in the quinary Nb-Mo-Ta-W-V space and trained a neural network model for barrier prediction. Figure R11 demonstrates the testing performance on five quaternary alloys (NbMoTaW, NbMoTaV, NbMoWV, NaTaWV, MoTaWV) and one quinary system NaMoTaWV. These atomic configurations were not part of the training and validation set. Our initial attempt

yields a mean absolute error of approximately 0.052 eV (within 4% of the mean barrier) for these alloys. Exploring the extensive range of compositional possibilities in CCAs is an intriguing topic that requires a highly scalable model and deserves further, meticulous study.

We thank for the reviewer's suggestion on using the recently developed machine learning interatomic potentials (MACE). It is encouraging to see that by employing higher-order body messages, the new MACE shows a fast and highly parallelizable performance, exceeding state-of-the-art accuracy. For the extended defects such as dislocation and grain boundary, using DFT will become computationally prohibited, and we are keen to explore the newly developed ML potentials.

Figure R11. Preliminary results show the scalability and performance of our model with more complex alloys. The ground truth values are DFT-NEB calculations of vacancy diffusion barriers. The left shows the quaternary systems, and the right is for quinary alloy.

C) In theory by expanding the conv net and including interstitial points the model has the potential to cover off-lattice transformations, could this be tested?

Reply: The reviewer raised an interesting question regarding the possible expansion of the model to other (off-lattice) defects. Our current research primarily concentrates on addressing the challenge of modeling vacancy diffusion and the resulting chemically ordered structures in compositionally complex materials. In response to your query, we will discuss the potential approaches for examining off-lattice defects, specifically interstitial and dislocation. A pivotal aspect is to represent these off-lattice defects using on-lattice neural map.

a. Interstitial defect. An interstitial defect is characterized by the presence of an additional atom occupying the interstitial site. In Figure R12a, we schematically show an interstitial atom within a square lattice. In terms of the neural map representation, the presence of an interstitial introduces an additional digit, indicating both the location and the type of the interstitial atom. This alteration in the neuron sequence is depicted in Figure R12c. The machine learning models, such as neural

networks, should be capable of recognizing these altered sequences and correlating them with its properties (diffusion barrier). Similar to vacancy diffusion, an interstitial also possesses multiple pathways. The rotational non-invariance can be utilized to predict path-dependent barriers effectively.

Figure R12. Chemical and structural representation of interstitial. (a) Crystal lattice containing an interstitial atom. (b) Neural map (digit matrix) representation of the configuration. (c) The presence of interstitial (neuron) alters the sequence of the neural map.

b. Dislocation defect. In a crystal containing a dislocation, atoms shift from their optimal lattice sites, resulting in a distortion (displacement) in the crystalline matrix. To capture the chemistry and structure of screw dislocation, we can incorporate two crucial inputs, the local atomic type (Figure R13a) and displacement (Figure R13c). Figure R13b shows its neuron map representation of local chemical environment of a screw dislocation. Correspondingly, the atomic displacement is encoded into a numerical matrix, in which each number is the precise atomic displacement due to the presence of dislocation (Figure R13d). In Figure R14, we present the preliminary results testing the performance, where predicted dislocation barriers are compared with NEB calculations.

Figure R13. Chemical and structural representation of screw dislocation. (a) Atomic structure containing a screw dislocation and its local chemistry. Atoms are colored-coded according to their type. (b) The corresponding chemistry map, where each digit denotes a specific type of atom (1, 2, and 3 represent Nb, Mo, and Ta, respectively). (c) The displacement field associated with the screw dislocation. Atoms are colored by their displacement. (d) The corresponding displacement map fully captures atomic displacement information.

Figure R14. Testing results showing the dislocation barrier prediction. Barrier spectra for alloys of (a) dilute and (b) concentrated solutions. The testing shows the predicted barriers from neural network model and true barriers. Both MAE and barrier distributions for each composition are shown.

D) The citations only seem to cover almost entirely famous high-impact publications but it seems like the authors are not fully caught up in the literature in techniques of mapping atomic configurations to properties.

Reply: These publications have motivated us to develop the model for revealing vacancy diffusion and new phase formation. We very much appreciate the reviewer's suggestion to incorporate more relevant literature. As a result, we have enhanced our discussion regarding the mapping of atomic configurations to properties, cluster expansion, machine learning models, et al. We have added 11 new references to the revised manuscript.

Minor Comments

D) They say that atomic potentials can't be 'directly' trained on diffusion barriers but this seems confusing and misleading. They are trained on the relevant input structures and compute the energy difference in the same way DFT would. Linear regression could be 'trained directly' but that is somewhat meaningless to mention. They should clarify the advantages of this 'direct' training.

Reply: For an initial state (local energy minimum), we meant to 'directly' predict its corresponding transition states and the associated energy barriers merely from this given minimum. However, we also agree with the reviewer that, the energy of a transition state can be trained based on the provided input structure. It's important to note that during the prediction phase, the configuration of the transition state is typically unknown. Consequently, accurately predicting the energy of a transition state requires first determining its structure, which might necessitate additional saddle point calculations. The proposed method focuses on a one-step process, where we predict diffusion energy barriers starting from an initial minimum state. To eliminate any confusion and possibly misleading information, we have made appropriate revisions to the sentence in the amended manuscript.

E) the use of jet colour scheme should be avoided as it give a false impression of gradients, the viridis colorscheme is more accurate to the eye [and is more accessible in black and white or for colorblind readers]

Reply: We have updated Figure 2 using the Viridis colormap in the revised manuscript, as we agree figures should be accessible and readable for all, including colorblind individuals. The reviewer raised an essential issue that future guidelines on scientific papers to enhance accessibility and readability could be implemented within the scientific community.

F) There is a growing body of literature using 3D conv nets, e.g., in MRI medical imaging, it would be nice if the authors made at least some small effort to establish what is occurring in the field.

Reply: Due to the exceptional capability to identify features and patterns in images, CNNs have been widely utilized in the field of image processing and analysis. In the context of MRI imaging, CNN-driven deep learning (DL) is transforming medical diagnostics, prognostics, and research.

To the best of the authors' knowledge, this includes applications in both accelerated acquisition and enhanced diagnosis.

Accelerated MRI acquisition. A challenge in MRI is the extensive scan acquisition time, which usually ranges from 20 to 40 minutes per imaged area. Deep CNN has been used to enhance the quality and decrease the time required for acquiring MRI medical images.

Detection and segmentation. The process of manual segmentation not only demands specialized expertise but also consumes a considerable amount of time. Automating this task, therefore, has significant value. In MRI, specifically in the segmentation of the myocardium, the performance achieved through DL now aligns closely with the variations typically seen between human experts (*IEEE transactions on medical imaging 37.11 (2018): 2514-2525*).

The reviewer may find the three review articles helpful, (1) *Nat. Rev. Rheumatol 18.2 (2022): 112-121*; (2) *Nat. Rev. Cardiol. 18.8 (2021): 600-609*, and (3) *Nat. Mach. Intell. 2.12 (2020): 737-748*, which nicely summarize the application of DL and AI in the MRI field.

G) The introduction says that current descriptors scale quadratically with the number of elements, they should look at recent papers by the Ceriotti [<https://journals.aps.org/prmaterials/abstract/10.1103/PhysRevMaterials.7.045802>] and Csányi [<https://journals.aps.org/prl/abstract/10.1103/PhysRevLett.131.028001>] who have made great progress in reducing the scaling with number of elements.

Reply: We thank the reviewer for providing the two references. It is great to see that researchers have recently developed new methods (tensor-reduced representation) to tackle the steep scaling in the size of representation as the number of elements grows. These references have been incorporated in the amended manuscript with the aim of informing future readers as well.

Reviewer #3:

This NNK framework will be an excellent tool in the area of CCAs to explore properties for vast compositional spaces. This manuscript should be published before fixing some issues as illustrated below

Reply: We thank the reviewer very much for the positive comments on the tool developed to explore the vast compositional space.

1. In the manuscript it is clearly mentioned how to generate the neuron map. However, why not use a graph neural network directly instead of creating the neuron map? Please add justifications for creating neuron maps over graph neural network-based approaches.

Reply: The motivation of using a neural map originates from its simplicity to mirror vacancy jumps through the swapping of neurons (digits). With only one-time conversion of atomic configuration to neuron map (Figure R15a), vacancy jumps and chemical evolution are simulated by swapping two digits (0 and one of its nearest neighboring neurons). In this way, millions of vacancy jumps can be modeled efficiently, with each jump iteration involving the action of just two neurons (illustrated in Figure R15b).

Graph neural network could theoretically predict vector properties using rotation-covariance features. Simulating vacancy diffusion using graph neural network entails node swapping and updating edge properties. Each node swap (vacancy jump) potentially affects the neighboring nodes and their connected edge features, necessitating their updates. This requires altering a significant portion of the network (possibly up to the 8th nearest neighbors of vacancy node) after each its jump, thus increasing computational cost. In contrast, the neural map approach only requires updating two neurons for each diffusion jump. Moreover, the application of graph neural network in path-dependent diffusion barrier prediction (a vector property) has yet to be developed and incorporating rotation-covariance is possibly nontrivial.

Figure R15. Vacancy jumps are mirrored by swapping of neurons. (a) Conversion of atomic structure to neuron map is required only once. (b) Vacancy jumps are modeled by swapping two digits (0 and one of its nearest neighboring neurons). Millions of vacancy jumps can be simulated efficiently, with each jump iteration involving the action of just two neurons.

2. The link under Data Availability is not the correct link.

Reply: Thanks to the reviewer. We have verified the DOI link to the raw data and can confirm that the URL <https://zenodo.org/records/7714651> (DOI: [10.5281/zenodo.7714650](https://doi.org/10.5281/zenodo.7714650)) is now working.

Revision and changes to the manuscript and SI

Major revisions

- Expanded the review of literature on techniques for mapping atomic configurations to properties (page 3 of the manuscript).
- Compared the cluster expansion method and discussed its challenges in predicting diffusion barriers in CCAs (page 12 of the manuscript).
- Discussed the use of graph neural networks and highlighted potential issues in modeling vacancy jumps (page 13 of manuscript).
- Highlighted the unique feature of NNK in mirroring vacancy jumps and chemical evolution (page 15 of manuscript).
- Cited 11 new reference papers to enrich the paper's bibliography.
- SI: Added a new Figure S1 to illustrate the transition state and diffusion barrier
- SI: Included data to interpret the rotational non-invariance of the neural map with new Figures S12 and S13 (page 12).
- SI: Added cross-validation data (page 15).
- SI: Introduced a new section, Section 6, which provides an extensive comparison with the cluster expansion method.
- SI: Included new Tables S2 and S3 to present the complete dataset.

Minor revisions

- Updated the color of Figure 2c for improved accessibility.
- The composition and dataset are clearly described in the Method section.
- Corrected the link to access all the data.
- SI: Updated the caption for Figure S2.
- SI: Clearly described the data used to validate the model (page 20 of SI).

References

1. X. Zhang, M. H. F. Sluiter, Cluster Expansions for Thermodynamics and Kinetics of Multicomponent Alloys. *J Phase Equilibria Diffus* **37** (2016).
2. Z. Xi, M. Zhang, L. G. Hector, A. Misra, L. Qi, Mechanism of local lattice distortion effects on vacancy migration barriers in fcc alloys. *Phys Rev Mater* **6** (2022).

REVIEWERS' COMMENTS

Reviewer #2 (Remarks to the Author):

I found it very strange the authors insist that 4 layers is 'converged' it seems very clear from their own data that two layers is converged and after it is neutral or overfitting. See R4 (b) that validation error (the one that 'really' counts) goes up after 2 layers, though only by a small amount. Training error drops by an equally small amount. The authors seem to have used more layers than they need, this is fine, I do not think they need to redo any work, but if this is going to form as a foundational text for the community we must be honest about what are good parameters (indeed they may be doubling the computational cost for no reason, perhaps even making the results worse).

I would personally be interested to see 2CNN and one FNN or even one CNN and one FNN. I don't think this is necessary but I think the author should be a bit more clear.

Regarding the Responses to Reviewer #2, I find the response "It has been noted that it can take a number of weeks to months to select optimal clusters" a bit strange. Weeks to months on what? What were the settings, on a desktop computer, on a cluster? This is not a very clear answer. If the algorithm does not parallelize this would be important to mention

I am very pleased with the detailed discussion the authors provide regarding CE, the authors clearly have now given careful attention to the issue.

I think the authors have done something very impressive in that they have an ML technique that is operating at 'classical' speeds, (actually faster it seems). This is an important result that should be very clearly emphasized early and often, most readers familiar with machine learning potentials will automatically assume this might be a slow technique.

10 million diffusion jumps on 128,000 atoms on a single cpu over 2 days is a very very nice result. I would encourage the authors to compare the costs of their descriptor relative to those used by MLP, which will be more accurate, but will not come anywhere close to the benchmarks achieved here.

If I understand from figure R11 the authors lose no accuracy with additional elements? This is an amazing result and I believe they should include it in the text.

I strongly encourage the authors to place their answers as an extended section in the appendix or supplementary materials. They have done some interesting preliminary work on interstitials and dislocations that could be quite interesting to the broader community.

It might be interesting to use other standard techniques, e.g, pooling layers to make a more sophisticated scheme that can better take in long-range effects (but that is beyond the scope of the work here, just a speculation).

Reviewer #3 (Remarks to the Author):

The responses to the queries are satisfactory to me and the manuscript can be published.

Reviewer #4 (Remarks to the Author):

What are the noteworthy results?

Novel use of CNN to run diffusion. Use of an aligning step allows for a clear and easy way to train for kinetics.

Will the work be of significance to the field and related fields?
Yes.

How does it compare to the established literature?
Compares well

If the work is not original, please provide relevant references.
Does the work support the conclusions and claims, or is additional evidence needed?
Sufficiently original

Are there any flaws in the data analysis, interpretation and conclusions? Do these prohibit publication or require revision?
No

Is the methodology sound? Does the work meet the expected standards in your field?
Yes

Is there enough detail provided in the methods for the work to be reproduced?
Yes

Based on the authors corrections I now recommend this be published.

I co-reviewed this manuscript with one of the other reviewers.

A point-by-point response to the reviewers' comments

Reviewer #2:

I found it very strange the authors insist that 4 layers is 'converged' it seems very clear from their own data that two layers is converged and after it is neutral or overfitting. See R4 (b) that validation error (the one that 'really' counts) goes up after 2 layers, though only by a small amount. Training error drops by an equally small amount. The authors seem to have used more layers than they need, this is fine, I do not think they need to redo any work, but if this is going to form as a foundational text for the community we must be honest about what are good parameters (indeed they may be doubling the computational cost for no reason, perhaps even making the results worse). I would personally be interested to see 2CNN and one FNN or even one CNN and one FNN. I don't think this is necessary but I think the author should be a bit more clear.

Reply: Thanks for the comments on the neural network architecture. We don't disagree with the reviewer. Indeed, as outlined in Section 4 of the Supplementary Information, it is acknowledged that "*The prediction error decreases with either increasing the number of layers or neurons and begin to converge for the model with 128 neurons and 2 layers*". The selection of a model with 4 layers is owing its robustness in equimolar concentration alloys, in which we elucidate the diffusion multiplicity and B2 ordering (i.e., the focus of our current study).

We have made our framework and source code publicly accessible, along with all the training data. This enables researchers to train various neural network kinetics models, using CNN or FNN, according to their research needs. For dilute alloys, a simpler version of model with 2 layers suffices to accurately predict all barriers and the diffusion-mediated ordering behaviors. Should any researchers encounter any issues or have questions regarding building the model for their problems using our codes, we warmly invite them to contact us without hesitation.

Regarding the Responses to Reviewer #2, I find the response "It has been noted that it can take a number of weeks to months to select optimal clusters" a bit strange. Weeks to months on what? What were the settings, on a desktop computer, on a cluster? This is not a very clear answer. If the algorithm does not parallelize this would be important to mention

Reply: The information is quoted from the literature (i.e., reference 4 in Supplementary Information). The process of designing clusters in Cluster Expansion (CE) is known to be time-consuming. With the introduced neuron map representation, the calculation becomes straightforward. In Section 6 of the Supplementary Information, we present a detailed comparison of our method with the cluster expansion approach.

I am very pleased with the detailed discussion the authors provide regarding CE, the authors clearly have now given careful attention to the issue.

Reply: We are pleased to know this.

I think the authors have done something very impressive in that they have an ML technique that is operating at 'classical' speeds, (actually faster it seems). This is an important result that should be very clearly emphasized early and often, most readers familiar with machine learning

potentials will automatically assume this might be a slow technique. 10 million diffusion jumps on 128,000 atoms on a single cpu over 2 days is a very very nice result. I would encourage the authors to compare the costs of their descriptor relative to those used by MLP, which will be more accurate, but will not come anywhere close to the benchmarks achieved here.

Reply: We thank reviewer 2 for the suggestion and acknowledge the efficiency of our neural network kinetics. In comparison to the descriptor of the MLP potential, our neuron map (on-lattice) representation, which fully captures the chemistry and structure of crystals, has an ultra-small feature size $O(n)$. Additionally, our framework, trained by small models, can be applied directly to investigate diffusion in large systems without sacrificing accuracy. This vital feature is underscored at various places within both the Manuscript and Supplementary Information, notably in the "Neural Network Kinetics Scheme" under the Results Section, the first two paragraphs of the Discussion section, and Sections 2 and 6 of the Supplementary Information.

If I understand from figure R11 the authors lose no accuracy with additional elements? This is an amazing result and I believe they should include it in the text.

Reply: The accuracy with increasing elements slightly decreases, given the small amount of training data we currently derive from density functional theory calculations. The performance of the neural network model with the chemical complexity (4, 5, ... 9 elements) presents an intriguing subject, meriting its own dedicated investigation, which is our next focus. We support publishing the Peer Review File, where all reviewers' questions, our answers, and preliminary data for other subjects will be accessible.

I strongly encourage the authors to place their answers as an extended section in the appendix or supplementary materials. They have done some interesting preliminary work on interstitials and dislocations that could be quite interesting to the broader community.

Reply: We recommend publishing our answers to the reviewers in the Peer Review File, as the Supplementary Information already encompasses all information pertinent to this study.

It might be interesting to use other standard techniques, e.g, pooling layers to make a more sophisticated scheme that can better take in long-range effects (but that is beyond the scope of the work here, just a speculation).

Reply: We agree with the reviewer's perspective that incorporating additional techniques is essential for addressing distinct challenges, including those related to long-range Coulomb interactions.

Reviewer #3:

The responses to the queries are satisfactory to me and the manuscript can be published.

Reply: We thank the reviewer for reviewing our revised manuscript and supporting its publication.

Reviewer #4:

What are the noteworthy results?

Novel use of CNN to run diffusion. Use of a aligning step allows for a clear and easy way to train for kinetics.

Will the work be of significance to the field and related fields?

Yes.

How does it compare to the established literature?

Compares well

If the work is not original, please provide relevant references.

Does the work support the conclusions and claims, or is additional evidence needed?

Sufficiently original

Are there any flaws in the data analysis, interpretation and conclusions? Do these prohibit publication or require revision?

No

Is the methodology sound? Does the work meet the expected standards in your field?

Yes

Is there enough detail provided in the methods for the work to be reproduced?

Yes

Based on the authors corrections I now recommend this be published. I co-reviewed this manuscript with one of the other reviewers.

Reply: We would like to thank reviewer #4 very much for reviewing our manuscript and our responses to reviewer 1, as well as for the recommendation of publication.